# Health-Related Quality of Life and Perceived Burden of Informal Caregivers of Patients with Rare Diseases in Selected European Countries

**DOI:** 10.3390/ijerph19138208

**Published:** 2022-07-05

**Authors:** Cristina Valcárcel-Nazco, Yolanda Ramallo-Fariña, Renata Linertová, Juan Manuel Ramos-Goñi, Lidia García-Pérez, Pedro Serrano-Aguilar

**Affiliations:** 1Fundación Canaria Instituto de Investigación Sanitaria de Canarias (FIISC), 38109 Santa Cruz de Tenerife, Spain; cristina.valcarcelnazco@sescs.es (C.V.-N.); yolanda.ramallofarina@sescs.es (Y.R.-F.); lidia.garciaperez@sescs.es (L.G.-P.); 2Network for Research on Chronicity, Primary Care, and Health Promotion (RICAPPS), 28029 Madrid, Spain; pseragu@gobiernodecanarias.org; 3Research Network on Health Services in Chronic Diseases (REDISSEC), 28029 Madrid, Spain; juanmanuel.ramosgoni@gmail.com; 4EuroQol Research Foundation, 3068 AV Rotterdam, The Netherlands; 5Servicio de Evaluación del Servicio Canario de la Salud (SESCS), 38109 Santa Cruz de Tenerife, Spain

**Keywords:** health-related quality of life, EQ-5D-5L, caregivers, rare diseases, informal care, caregiver burden

## Abstract

Most of rare disease (RD) patients are assisted in their homes by their family as informal caregivers, causing a substantial burden among family members devoted to care. The role of informal caregivers has been associated with increased levels of stress, poor physical/mental health and impaired HRQOL. The present study assessed the impact on HRQOL and perceived burden of long-term informal caregiving, as well as the inter-relationships of individuals affected by different RD in six European countries, taking advantage of the data provided by the BURQOL-RD project (France, Germany, Italy, Spain, Sweden and UK). Correlation analysis was used to explore the relation between caregiver HRQOL and caregiver burden (Zarit Burden Interview). Multinomial logistic regression models were used to explore the role of explanatory variables on each domain of caregivers HRQOL measured by EQ-5D. Caregivers’ HRQOL is inversely correlated with burden of caring. Mobility dimension of EQ-5D was significantly associated with patients age, time devoted to care by secondary caregivers, patient gender and patient utility index. Patients’ age, burden scores and patient utility index significantly predict the capacity of caregivers to perform activities of daily living. Employed caregivers are less likely of reporting ‘slight problems’ in pain/discomfort dimensions than unemployed caregivers. The EQ-5D instrument is sensitive to measure differences in HRQOL between caregivers with different levels of burden of care.

## 1. Introduction

Although rare diseases (RDs) are characterized by a low incidence (5/10,000 persons) and prevalence, over 30 million EU citizens suffer from these conditions, given the existence of more than 5000 RDs [1]. In most cases, RDs are chronic and debilitating conditions with no effective therapy, requiring long-term and costly formal and informal care [1]. Up to 25% of patients with RDs waited five years or more for a diagnosis and almost 40% received an incorrect initial diagnosis [2]. These facts, together with the limited availability of effective treatment alternatives, often underly the shortened life expectancy and quality of life for these patients [3].

Most RD patients are assisted in their homes by their family as informal caregivers, causing a substantial burden among family members devoted to care [4]. Caregiver burden is a multidimensional response to physical, psychological, emotional and social stressors associated with the caregiving experience [5]. Differences have been reported for family caregivers regarding the burden of caring and health related quality of life (HRQOL) according to the severity of the RDs [6,7].

The role of informal caregivers has been associated with increased levels of stress, poorer physical/mental health and impaired HRQOL [8]. The relationship between caregiver’s HRQOL and the perceived burden has been studied for common chronic diseases [9,10,11], but not for RDs, reporting a clear negative relationship between the burden of caring and HRQOL, measured with the aid of generic HRQOL measurement instruments such as EQ-5D [12] or SF-36 [13]. Although some country-specific research on a limited set of RD has been done on HRQOL and caregivers burden [4,5,6,7], cross-national comparative research on these issues is still scarce. EQ-5D vas scores and mean EQ-5D index scores are clearly reduced, both for patients affected by RD such as Duchenne muscular dystrophy [7] and their caregivers [5]. However, the relationship between the burden of care for family caregivers and the self-perceived HRQOL needs to be better understood given that recently reported differences in burden of care between family caregivers providing full home care and family caregivers supported by formal or institutional care were not followed by differences in HRQOL [14]. Information on HRQOL helps to define the global impact of a specific health problem on society and is increasingly used as a useful indicator together with economical and epidemiological data to set priorities and allocate health care and social resources [15,16].

‘Social economic burden and health-related quality of life in patients with rare diseases in Europe’ (BURQOL-RD project) was a three-year project under the second Programme of Community Action in the Field of Public Health, which has implemented tools to measure the current impact of 10 RDs on socioeconomic issues and HRQOL within the EU [17]. The outcomes provided by the BURQOL-RD project approach and their application in person-centered assessments and decision making is relevant to define the current consequences of rare diseases policies on society, and to assess the effectiveness of new policies and specific interventions [17]. The BURQOL-RD project included measurement instruments for family caregivers such as the Zarit Burden Interview (ZBI) for the burden of caring and the EQ-5D-5L, allowing the examination of their potential relationships and the identification of determinants of HRQOL among caregivers of RDs patients in different EU countries. The BURQOL-RD data makes possible the examination of the relationships between caregivers and patients characteristics and their determinants on a broad group of RDs affecting adults and children in several European countries.

The aim of this study was to examine the HRQOL difference between caregivers with different levels of caregiving burden, and to explore the factors associated with HRQOL in informal caregivers of patients affected by different types of rare diseases in selected European countries.

## 2. Materials and Methods

### 2.1. Study Design and Data Collection

BURQOL-RD project is a European project whose main objective was to develop a disease-based model capable of quantifying the economic burden and HRQOL for patients with RDs (Cystic Fibrosis, Prader-Willi Syndrome, Fragile X Syndrome, Hemophilia, Duchenne Muscular Dystrophy, Epidermolysis Bullosa, Scleroderma, Mucopolysaccharidosis, Juvenile Idiopathic Arthritis and Histiocytosis), and for their caregivers, from a societal perspective in eight European countries: Bulgaria, France, Germany, Hungary, Italy, Spain, Sweden and the UK [17]. This paper analyses a part of the dataset gathered within BURQOL-RD project, related to HRQOL of informal caregivers. This project was exempted by local ethics committees and all participants gave their informed consent to participate.

The fieldwork was carried out between September 2011 and April 2013. Once the translation and adaptation to each language was finished, a set of questionnaires were created in an online and paper version. The questionnaires were totally anonymous, as the patients or their caregivers were contacted by their patient organization or registry by email, and the responses were not associated with any identifying data (name, ID, address, e-mail). Completed questionnaires were automatically saved in a centralized database. A small part of the patients/caregivers who could not be contacted via email was provided with a paper version of the questionnaire with a prepaid envelope to return their response to the researchers. A link to the online questionnaire was also published in specialized social media profiles and web sites. Each questionnaire gathered data on patient’s burden, and his/her informal caregiver’s burden, if the patient had one.

In the present study, participants were informal caregivers who provided care on a daily basis to a person diagnosed with one of the 10 selected RDs from six of eight countries included in the BURQOL-RD project: France (N = 147 caregivers), Germany (N = 124 caregivers), Italy (N = 229 caregivers), Spain (N = 221 caregivers), Sweden (N = 28 caregivers) and the UK (N = 76 caregivers). Bulgaria and Hungary were excluded from the analysis because of the insufficient number of valid questionnaires to fulfill our main objective.

### 2.2. Measures

#### 2.2.1. Caregiver Burden and Informal Care

Caregiver burden was measured using the 22-item ZBI [18] validated for every participant country [19]. For each item, caregivers were asked to indicate how often they felt a particular way in a specific situation: never, rarely, sometimes, quite frequently or nearly always. Scores for each item ranged from 0 (never) to 4 (almost always). Total scores ranged from 0 to 88, with a higher score indicating greater burden [18]. This scale allows us to assess the level of caregiver claudication defined as the incapacity of a caregiver to provide the care required by the patient. A score less than 47 points in ZBI means the ‘absence of claudication’; a score between 47 and 55 points means ‘mild claudication’; and scores above 55 points imply ‘high level of claudication’ as usual [20].

Informal care, defined as the performance of tasks by non-professionals caregivers that help maintain or enhance the patient’s independence, was obtained from the questionnaires by means of specific items concerning the time spent helping the patient with his or her basic activities of daily living, and the time spent helping with necessary instrumental activities of daily living.

#### 2.2.2. Health-Related Quality of Life

Health-related quality of life was measured using EQ-5D [21] in adult patients and caregivers. The specific EQ-5D-Y version was used for children patients in all countries but France where EQ-5D was used because the French children version was not available.

The EQ-5D questionnaire [12] is a recognized generic instrument for measuring HRQOL that provides a descriptive profile and a single index value for health (index score). It has been deemed a valid cross-sectional, generic measure of health outcomes in rare diseases and is commonly used as the basis for economic evaluations. Existing literature report that EQ-5D might be insensitive to capture some of the salient impairment both in the patients’ HRQL in several conditions (e.g., breast and prostate cancer, asthma and myocardial infarction) and in caregivers of dementia patients (low sensitivity to change with time and poor correlation with self-perceived burden measured by the ZBI) [16].

The EQ-5D provides a descriptive profile and a single index value for health which was designed for self-completion by respondents. The used EQ-5D 5-level version (EQ-5D-5L) [22] comprises 5 dimensions: mobility, self-care, usual activities, pain/discomfort and anxiety/depression. Each dimension has 5 levels: no problems, slight problems, moderate problems, severe problems and unable/extreme problems.

EQ-5D-Y is a modified version of the EQ-5D for use in respondents from 8 years onwards which includes the advantages of the standard adult EQ-5D. The EQ-5D-Y is still conceptually equivalent to the EQ-5D and provides an opportunity to compare children’s and adult’s ratings in corresponding dimensions [23].

The EQ-5D utility index is a continuum measure where 1 represents best possible health and 0 represents death, with negative values for health states considered worse than death. The ability to convert self-classifier responses into a single preference-based score makes the EQ-5D practical for clinical and economic evaluation [24]. In our case, EQ-5D-5L were converted to a utility index by applying the algorithm derived from the crosswalk 3L value sets [25] which translate 5L responses into 3L responses and apply 3L value sets. For EQ-5D-Y, the utilities were derived from the US Measurement and Valuation of Health [26].

#### 2.2.3. Other Information

Satisfaction with the National Health System (NHS) was measured in a 1 to 10 scale where 1 means ‘unsatisfied’ and 10 means ‘very satisfied’.

Other information collected about the caregiver included age, sex, working situation of main caregiver and the family relationship with the patient. The working situation of the main caregiver was categorized as ‘employed’, ‘retired or pensioner’ and ‘housewife or househusband’. Familiar relationship with patient was categorized into ‘spouse’, ‘son or daughter’, ‘mother or father’ and ‘other’. For each patient, sex and age data were also recorded.

### 2.3. Statistical Analysis

Descriptive analysis was used to present sample characteristics. Means and standard deviations were used to describe continuous variables and frequencies were used to describe categorical variables. Correlation analysis by means of Pearson coefficient was used to explore the relation between caregiver HRQOL and caregiver burden (ZBI).

#### 2.3.1. The EQ-5D Capacity to Discriminate HRQOL between Caregivers’ Burden Groups

After checking the homogeneity of variances with Levene’s test, ANOVA analysis was used to evaluate differences in EQ-5D index (crosswalk values) between the caregivers group according to the caregiver’s burden defined by ZBI groups (absence of claudication, mild claudication and high level of claudication). The Bonferroni correction was used for multiple comparisons (P^c^, corrected P-value).

#### 2.3.2. Factors Related to Caregivers’ HRQOL

Regression models allow us to explain the caregivers’ EQ-5D health status through explanatory variables such as gender of caregiver, age of patient, ZBI score, time spent in providing care each day and the patient’s HRQOL. Although the response categories from the EQ-5D follow a natural ordering, an ordered logistic approach was not considered because the proportional odds assumption was not met, meaning, response levels are not equidistant. The multinomial logistic regression estimates simultaneous binary logistic models for all the possible comparisons among J categories [27,28]. First, we performed a regression model using caregivers EQ-5D responses for mobility dimension as a dependent variable by including all explanatory variables in the model and subsequently dropping and adding variables to come up with the best-fitted model, based on adjusted R2 values and Akaike information criterion (AIC). The explanatory variables included in first step were: patient age, satisfaction with the NHS, EQ-5D adult patients index, EQ-5D child patients index, care hours of main caregiver, care hours of other caregiver, years of caring, ZBI score, relationship with the patient and caregiver sex. Caregiver age was not included due to the high rate of missing values for this variable in the BURQOL-RD database. Once the best model was estimated, we repeated the same process for the rest of the EQ-5D dimensions. Categorical variables are introduced in the model as dummies variables. The effects on each domain of caregivers EQ-5D of explanatory variables were examined using five multinomial logistic regression models, one by each domain. Each of the EQ-5D dimensions can take any of the five possible values: ‘No problems’ (reference group), ‘slight problems’, ‘moderate problems’, ‘severe problems’ or ‘unable/extreme problems’. However, for modelling purposes, these last two categories were grouped due to insufficient sample size to include them separately.

Truncated multiple regression models were performed to analyze the effect of different variables on caregivers’ HRQOL scores. However, the models’ fit were poor and we decided to do not present it.

Statistical analysis and regression modelling were conducted in STATA MP 11 [29]. Level of significance of 0.05 was considered in the analysis.

## 3. Results

We obtained 825 caregivers data where both ZBI and caregivers EQ-5D questionnaires were available from BURQOL-RD database; Sweden and Italy has the highest proportion of males among the caregiver sample and UK had the lowest proportion of males, having the highest mean utility values among countries (Table 1).

There were 234 (28.4%) adult patients’ caregivers and 591 (71.6%) child patients’ caregivers in the sample. The majority of caregivers were women (80.2%), parents of the patients (49.1%) and employed (55.2%) (Table 2). The mean years of caring were 9.01 (sd: 9.46) and the mean hours of care per week was 54.65 (sd: 35.93) hours. A total of 84% of caregivers reported ‘absence of claudication’ according to ZBI questionnaire. Furthermore, in this study, 62.5% of patients were female and mean age were 14.93 (sd: 14.18) years.

The distribution of the EQ-5D response levels of caregivers by dimension shows how more than 20% of the caregivers self-reported moderate, severe or extreme problems in Pain/Discomfort or Anxiety/Depression dimensions (Table 3). However, for mobility and self-care, more than 80% of caregivers have no problems or slight problems.

### 3.1. The EQ-5D Capacity to Discriminate HRQOL between Caregivers’ Burden Groups

Results show that caregiver HRQOL is inversely correlated with its burden; the greater the caregiver burden, the lower the HRQOL (Pearson correlation coefficient = −0.180; *p*-value < 0.0001). Although a statistically significant correlation was obtained, the effect size was small. ANOVA analysis revealed that EQ-5D mean index are different between caregivers burden groups (Figure 1). When comparing the ‘absence of claudication’ group with the ‘mild claudication’ group and ‘high level of claudication’, the differences in the EQ-5D index were found to be statistically significant (P^c^ < 0.05 in both comparisons), although the differences in the EQ-5D index between the ‘mild claudication’ group and ‘high level of claudication’ were not statistically significant (P^c^ = 0.254).

### 3.2. Factors Related to Caregivers’ HRQOL

The Mobility dimension was significantly associated with patient age, care hours per week from other caregiver who is not the main caregiver, patient sex and patient utility index. Thereby, the main protective factors for mobility dimension were caring for a female and younger patient (Table 4). In the Self-Care dimension, caregivers who perform between 0 to 5 years of caring are 2.7 times more likely of having slight problems. This probability is about 3.9 times for caregivers who perform between 5 to 10 years of caring. In this case, caregivers are 3.4 times more likely of having severe or extreme problems (Table 4). In this way, taking care of the patient for less than 10 years increased the risk of having self-care problems. ZBI, caregiver sex, patient sex and patient utility had a significant association with this EQ-5D dimension. In relation to the Usual Activities dimension, patient age, ZBI and patient utility significantly predict the level of problems in performing caregiver’s usual activities. The working situation of the caregiver was significantly associated with the Pain/Discomfort dimension. Caregivers employed were significantly less likely of having slight pain/discomfort than unemployed caregivers. Furthermore, caring younger patients and with a higher utility index were a protective factor for Pain/Discomfort (Table 4). Furthermore, patient age and patient utility explain differences in the Pain/Discomfort dimension.

Satisfaction with NHS, ZBI and the absence of claudication (ZBI) were protective factor for the Anxiety/Depression dimension of caregivers (Table 4). Female caregivers are 2.1 times more likely of having slight problems related to anxiety or depression than male caregivers.

Care hours per week of main caregiver and relationship with the patient did not significantly influence the EQ-5D dimensions.

## 4. Discussion

This study assessed the impact on HRQOL and perceived burden of informal caregiving, as well as the inter-relationshipsof individuals affected by different RD in six European countries, taking advantage of the data provided by the BURQOL-RD project dataset [17]. This research article examines the impact of informal caregiving on self-reported HRQOL and perceived burden, as well as their inter-relationships and determinants, cross-nationally, among individuals affected by different RD, in six European countries, taking advantage of the data provided by the BURQOL-RD project dataset [17]. The BURQOL-RD project was a 3-year European project funded by the Directorate General for Health and Consumer Affairs (DG Sanco) [17], with the main objective of developing a disease-based model capable of quantifying the socio-economic burden and HRQOL of patients/families suffering from ten rare diseases and their caregivers in European countries, setting the basis for an integrated and harmonized approach to assess public policies and interventions for rare diseases in the EU. The aim of this publication is to focus on the HRQOL difference between caregivers with different levels of caregiving burden, and to explore the factors associated with HRQOL among caregivers of patients affected by different types of RDs. As in most studies reporting on caregiving, women and parents of the patients assumed the majority of the informal care. The caregiver burden quantified by the ZBI showed that 84% of caregivers reporting “absence of claudication” had a 0.74 EQ-5D index (UK values).

We provide some evidence that indicates that the EQ-5D-5L instrument is sensitive to differences in HRQOL between caregivers with different burdens of care. This result highlights the potential of using this instrument to capture wider effects in HRQOL which incur not only to the patients under study, but also among their caregivers. Patient age and HRQOL as well as caregivers self-perceived burden of care were significant determinants for the level of problems in most EQ-5D dimensions of caregivers. The level of self-perceived burden by caregivers had a significant relation with the Anxiety/Depression dimension of EQ-5D, highlighting the significant pressure of caring imposed over informal caregivers. This finding has been previously reported in the literature on caregiving, calling for the need of making available psychological services and social support interventions to alleviate the psychological status among informal caregivers [5,13].

The relationship between caregiver’s HRQOL and the perceived burden has been extensively studied in common chronic diseases such as cerebrovascular disease and stroke [11,30], diabetes [11], dementia [12,14], cancer [10] and Parkinson disease [9]. Less frequent information on the impact of caregiving burden in caregivers HRQOL is available for RDs [4,13]. The SF-36 was used to describe the impact of self-perceived burden of caring in family caregivers of patients with Multiple Sclerosis (MS), reporting that significant predictors of burden were Role-emotional Functioning and Vitality dimensions, SF-36 scores of caregivers and the Expanded Disability Status Scale scores [13]. The CarerQol questionnaire was also used to describe the impact of self-perceived burden of caring in family caregivers of patients with Pompe disease [4], reporting that higher disease burden was associated with length of informal care. Half of the informal caregivers reported mental health problems and problems with daily activities. We have not found previous reports on the relationships between the burden of caring and HRQOL, in the context of RDs by means of the EQ-5D [5,6,7].

The international literature clearly states that caring for patients with rare diseases is burdensome and related to HRQOL, but interpretation of this relationships is lacking. Our report provides new information to interpret the relation between burden of caring and HRQOL in the context of rare diseases. Despite the scarcity of literature relating HRQOL and the self-perceived burden of care by caregivers of people affected by rare diseases, the available results seem quite consistent, regardless of the instrument used. According to our results, supported by Crossnohere et al. for Duchenne Muscular Dystrophy [31], we found support for the appropriateness of EQ-5D to assess caregivers self-perceived health status by level of burden of caring for several rare diseases in European countries. Whereas other specific measures may be more sensitive to specific outcomes in other rare diseases such as Amyotrophic Lateral Sclerosis, the EQ-5D provides the additional value of estimating health utilities for economic evaluations on potential interventions [31]. Similar to our results reported here, but measured by the McGilll questionnaire, the caregiver’s HRQOL and burden, and more specifically the weekly caregiving duties, age and health of the caregiver [32]. In addition, using the Medical Outcomes Short Form 36 questionnaire in mothers of children with mitochondrial disease, a significantly higher caregiver burden and poorer HRQOL, particularly related to role limitations, vitality and mental health were observed [33].

This study underlines the importance of also considering caregivers in the context of economic evaluations. It indicates that general HRQOL measures, as used in patients, may be able to detect HRQOL effects in caregivers, which facilitates the incorporation in common economic evaluations of HRQOL effects in care. Analysts and policy makers should be aware that if HRQOL improvement is an important aim, they should register HRQOL changes not only in patients but also in their caregivers.

For health policy assessment, monitoring and priority-setting, as well as for health and social care interventions evaluationand patient organizations’ decision on health and social resources allocation, HRQOL and the burden of disease have been found to be valuable self-reported measures, contributing to person-centered care. Combined with other clinical and economic evidence, the assessment and monitoring process is more informative, robust and valid. Available results illustrate that mean EQ-5D index scores for RD patients and caregivers are substantially lower than the equivalent in the general population, indicating lower quality of life. Several studies have shown that the provision of informal care is associated with serious adverse health effects for the caregiver, including anxiety and depression, impaired immune system function and coronary heart disease, as well as social isolation, financial deprivation and even premature death [8,9,10,11]. Most existing research, however, focuses on the burden on caregivers to elderly patients, predominantly with dementia, with less attention devoted to RD. Despite the descriptive nature of our data, observed results have several implications for health policy. First, given the association between number of hours spent providing informal care and caregiver burden, respite care and similar initiatives are urgently needed to help reduce the family burden and improve caregiver well-being. Second, our results suggest that many families caring for persons with RD require increased financial support to alleviate the considerable cost burden associated with RD. We must also emphasise the need to increase resources dedicated to social support for families (including reinforcement of formal care), and to improve visibility and social protection of informal caregivers, while also improving the social recognition of their work.

### Limitations

This study has some weaknesses. The presented results are a part of an extensive research on social economic burden and HRQOL (BURQOL-RD project). Therefore, one limitation of our study is that the BURQOL-RD database used was collected for other objectives and analysis. In this sense, a study limitation relates to the high rate of missing values for the caregiver’s age in the BURQOL-RD database that did not allow us to analyze it and related to their HRQOL. The age of caregiver is probably a relevant factor of the caregiver’s HRQOL; the older the caregiver, the poorer the HRQOL. However, we do not believe that adding this variable to our analysis would change the conclusions obtained. Furthermore, the specific EQ-5D-Y version was used for children patients in all countries but France where EQ-5D-5L was used in the children’s population. This may influence ratings of HRQOL obtained in this study. Child utility values were obtained from the Craig Valuation of child health-related quality of life in the United States study [26]. This value set study was performed by mean of a different methodology than the usual TTO studies for the 3L version, making the values to be on a different scale. However, it does not affect to the performed analysis as it is used just as explanatory variable in the regression models.

## 5. Conclusions

In conclusion, we show that the greater the caregiver burden, the lower the caregivers HRQOL is. Our findings suggest that poor patient utility index and high level of perceived care burden have a significant negative effect on mobility, self-care, usual activities and pain/discomfort of caregivers of RDs patients. From a clinical perspective, the knowledge generated in this study should be taken into account by professionals and authorities working in the field of RDs to identify existing problems and therefore, to design initiatives to better meet the needs of the RD patients, their families and caregivers.

## Figures and Tables

**Figure 1 ijerph-19-08208-f001:**
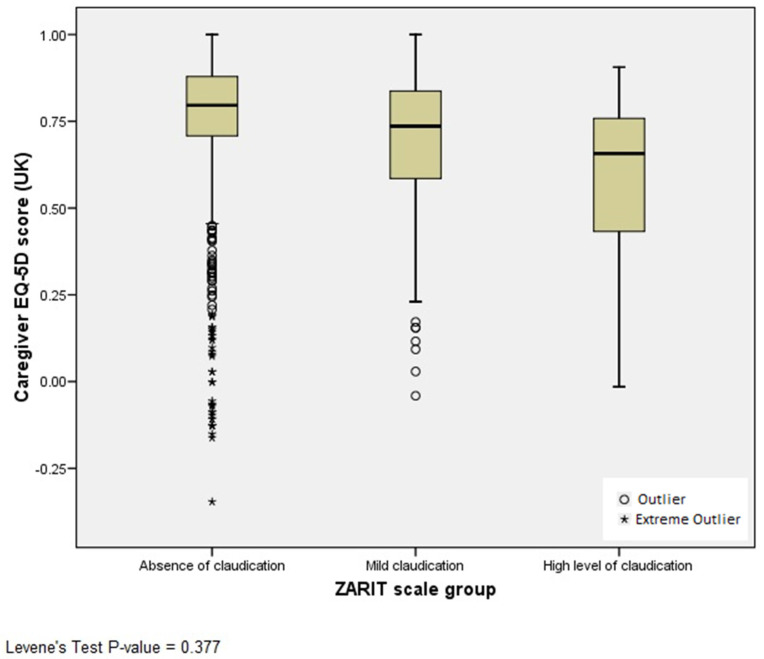
Box plot showing caregiver EQ-5D index (UK) of three groups of ZBI caregiver burden (Zarit scale group).

**Table 1 ijerph-19-08208-t001:** Characteristics of caregivers and patients by country.

	France (N = 147)		Germany (N = 124)		Italy (N = 229)		Spain (N = 221)		Sweden (N = 28)		UK (N = 76)	
	Caregivers	Patients	Caregivers	Patients	Caregivers	Patients	Caregivers	Patients	Caregivers	Patients	Caregivers	Patients
Female, N (%)	121 (82.3%)	53 (36.1%)	110 (88.7%)	59 (47.6%)	162 (70.7%)	73 (31.9%)	182 (82.4%)	79 (35.7%)	18 (64.3%)	14 (50%)	69 (90.8%)	30 (39.5%)
Age, mean ± sd	NA	22.83 ± 15.59	NA	30.48 ± 13.18	NA	35.54 ± 16.70	NA	31.08 ± 14.06	NA	34.14 ± 12.43	NA	31.30 ± 11.45
EQ-5D index (UK), mean ± sd	0.715 ± 0.256	0.499 ± 0.284	0.619 ± 0.344	0.235 ± 0.271	0.736 ± 0.255	0.225 ± 0.325	0.702 ± 0.257	0.335 ± 0.306	0.625 ± 0.314	0.451 ± 0.245	0.787 ± 0.240	0.239 ± 0.297
EQ-VAS, mean ± sd	74.35 ± 20.18	64.55 ± 20.62	74.71 ± 15.86	64.35 ± 21.71	76.67 ± 17.12	61.43 ± 21.16	71.89 ± 18.00	63.94 ± 21.00	72.30 ± 17.90	65.61 ± 15.67	77.37 ± 18.56	63.13 ± 19.01
ZBI, N (%)												
Absence of claudication	103 (70.1%)	--	115 (92.7%)	--	209 (91.3%)	--	179 (81%)	--	23 (82.1%)	--	64 (84.2%)	--
Mild claudication	29 (19.7%)	--	9 (7.3%)	--	16 (7.0%)	--	23 (10.4%)	--	4 (14.3%)	--	8 (10.5%)	--
High level of claudication	15 (10.2%)	--	0 (0%)	--	4 (1.7%)	--	19 (8.6%)	--	1 (3.6%)	--	4 (5.3%)	--

NA: Not available; sd: Standard deviation; ZBI: Zarit Burden Interview.

**Table 2 ijerph-19-08208-t002:** Characteristic of caregivers and patients (N = 825).

	Caregivers	Patients
Female, N (%)	662 (80.2%)	516 (62.5%)
Age, mean ± sd	NA	14.93 ± 14.18
EQ-5D index (UK), mean ± sd	0.737 ± 0.244	0.324 ± 0.313
EQ-5D index (Spain), mean ± sd	0.773 ± 0.268	0.290 ± 0.378
EQ-5D index (France), mean ± sd	0.735 ± 0.282	0.221 ± 0.312
EQ-VAS, mean ± sd	75.52 ± 17.79	63.00 ± 20.47
Years of caring, mean ± sd	9.01 ± 9.46	--
NHS Satisfaction, mean ± sd	--	6.54 ± 2.32
Relationship to patient, N (%)		
Spouse	166 (20.1%)	--
Son/Daughter	224 (27.2%)	--
Mother/Father	405 (49.1%)	--
Other	30 (3.6%)	--
Working situation, N (%)		
Employed	455 (55.1%)	--
Retired or pensioner	69 (8.4%)	--
House wife/House husband	241 (29.2%)	--
No reply	60 (7.3%)	--
ZBI, N (%)		
Absence of claudication	693 (84%)	--
Mild claudication	89 (10.8%)	--
High level of claudication	43 (5.2%)	--
Care hours per week (main carer), mean ± sd	54.65 ± 35.93	--
Care hours per week (other carers), mean ± sd	20.16 ± 0.28	--

sd: Standard deviation; NA: Not available; ZBI: Zarit Burden Interview.

**Table 3 ijerph-19-08208-t003:** Tabulations of caregivers EQ-5D response levels (Frequency, %).

Problem	Mobility	Self-Care	Usual Activities	Pain/Discomfort	Anxiety/Depression
No problems	643 (77.9%)	669 (81.1%)	518 (62.8%)	366 (44.4%)	324 (39.3%)
Slight problems	94 (11.4%)	62 (7.5%)	156 (18.9%)	276 (33.5%)	301 (36.5%)
Moderate problems	46 (5.6%)	44 (5.3%)	101 (12.2%)	144 (17.5%)	145 (17.6%)
Severe problems	21 (2.5%)	15 (1.8%)	35 (4.2%)	34 (4.1%)	40 (4.8%)
Extreme problems	21 (2.5%)	35 (4.2%)	15 (1.8%)	5 (0.6%)	15 (1.8%)

**Table 4 ijerph-19-08208-t004:** Odds ratio (Standard error) Multinomial logit regression model results for each dimension of EQ-5D-5L.

	Mobility			Self-Care			Usual Activities			Pain/Discomfort			Anxiety/Depression		
	Slight Problems	Moderate Problems	Severe Problems + Unable/Extreme Problems	Slight Problems	Moderate Problems	Severe Problems + Unable/Extreme Problems	Slight Problems	Moderate Problems	Severe Problems + Unable/Extreme Problems	Slight Problems	Moderate Problems	Severe Problems + Extreme Problems	Slight Problems	Moderate Problems	Severe Problems + Extreme Problems
**Patient age**	1.008 (0.010)	1.021 (0.132)	1.047 ** (0.150)	0.978 (0.015)	1.009 (0.130)	1.020 (0.160)	0.976 * (0.010)	1.009 (0.010)	1.007 (0.144)	0.999 (0.008)	1.023 * (0.010)	1.061 ** (0.015)	0.990 (0.008)	1.016 (0.010)	1.026 (0.137)
**Satisfaction with NHS**	0.993 (0.057)	1.046 (0.087)	1.108 (0.104)	1.029 (0.768)	1.014 (0.077)	1.031 (0.101)	0.989 (0.051)	1.010 (0.060)	1.061 (0.089)	0.959 (0.043)	0.969 (0.054)	1.066 (0.097)	0.995 (0.046)	0.915 (0.053)	0.848 * (0.065)
**Hours/week other caregiver**	1.007 (0.005)	0.999 (0.007)	1.016 * (0.006)	1.011 (0.006)	1.006 (0.006)	1.009 (0.007)	1.000 (0.004)	1.005 (0.005)	1.011 (0.005)	1.006 (0.004)	1.004 (0.005)	1.006 (0.008)	0.999 (0.004)	0.995 (0.005)	0.993 (0.007)
**Years of caring**															
0 to 5 years	1.221 (0.420)	1.800 (0.845)	2.044 (1.041)	2.701 * (1.285)	0.695 (0.327)	2.276 (1.284)	0.754 (0.231)	0.748 (0.270)	1.464 (0.680)	0.630 (0.166)	0.943 (0.298)	1.189 (0.632)	1.016 (0.268)	1.228 (0.397)	1.108 (0.516)
5 to 10 years	1.385 (0.469)	1.746 (0.875)	2.589 (1.289)	3.870 ** (1.740)	1.385 (0.588)	3.356 * (1.844)	1.303 (0.386)	1.637 (0.543)	1.434 (0.707)	0.683 (0.181)	0.957 (0.311)	1.389 (0.753)	0.961 (0.247)	0.661 (0.238)	0.682 (0.352)
**Caregiver working situation**															
Employed	0.845 (0.244)	0.643 (0.262)	0.750 (0.330)	1.427 (0.520)	0.849 (0.322)	1.039 (0.488)	1.080 (0.270)	1.565 (0.463)	0.696 (0.284)	0.615 * (0.135)	1.170 (0.314)	0.456 (0.214)	1.261 (0.279)	1.050 (0.297)	0.873 (0.345)
**ZBI**															
Absence of claudication	0.469 (0.218)	1.443 (1.533)	0.986 (0.881)	0.283 * (0.145)	0.899 (0.713)	0.627 (0.539)	0.376 * (0.162)	0.476 (0.267)	0.246 * (0.148)	1.268 (0.553)	0.602 (0.285)	0.307 (0.232)	0.259 * (0.170)	0.085 ** (0.059)	0.031 ** (0.022)
**Caregiver sex**															
Female	0.868 (0.302)	1.019 (0.450)	0.723 (0.340)	0.818 (0.348)	0.446 * (0.184)	0.515 (0.251)	1.296 (0.425)	0.884 (0.296)	0.629 (0.280)	1.376 (0.361)	1.729 (0.567)	1.446 (0.772)	2.107 ** (0.560)	1.310 (0.423)	1.688 (0.814)
**Patient sex**															
Female	0.303 ** (0.085)	0.955 (0.391)	1.531 (0.727)	0.492 * (0.173)	0.899 (0.342)	1.761 (0.920)	0.621 (0.151)	0.723 (0.211)	1.256 (0.539)	0.668 (0.147)	0.752 (0.204)	0.558 (0.249)	0.800 (0.174)	1.318 (0.387)	0.999 (0.397)
**Patient utility index**	0.818 (0.093)	0.854 (0.138)	0.579 ** (0.079)	1.062 (0.173)	0.857 (0.113)	0.713 * (0.101)	0.914 (0.092)	0.771 * (0.083)	0.774 (0.104)	0.784 * (0.087)	0.570 ** (0.067)	0.501 ** (0.088)	0.991 (0.091)	0.900 (0.099)	0.887 (0.143)
AIC	876.67			773.70			1151.69			1300.89			1319.27		
BIC	1057.46			954.48			1332.47			1481.68			1500.05		

NHS: National Health System; * *p*-value < 0.05; ** *p*-value < 0.01; ZBI: Zarit Burden Interview; AIC: Akaike information criteria; BIC: Bayesian information criteria.

## Data Availability

The data that support the findings of this study are available from BURQOL-RD project with restrictions, upon reasonable request to the authors and with permission of BURQOL-RD.

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
