# Peer review of "Health-Related Quality of Life and Perceived Burden of Informal Caregivers of Patients with Rare Diseases in Selected European Countries"

_ijerph, 2022, doi:10.3390/ijerph19138208_

Round 1
Reviewer 1 Report
The manuscript is well written, however I have comments that I believe the authors can improve further:
- Throughout the manuscript, I could not appreciate a proper definition or conceptualization of rare diseases (RD). It would be good if a conceptualization definition is provided and some examples on the prevalence rate or burden are reported in the introduction part.
- In the methods part, it is worthwhile to report some of the reliability parameters/diagnostic measures of scales or instruments used, especially for the population being studied. Comparisons with other literature would be helpful.
- Your reporting of decimals in results tables are inconsistent for mean (SD) scores, some up to two decimals, some four, etc. Please standardize them.
- For scores, please report the minimum and maximum values.
- The tables presented are a little difficult to read, I suggest to maximize the table size with a landscape view.
- Table 4 reports odds ratio and standard errors. It would be more appropriate to report their corresponding 95% confidence intervals.
- The discussion is relatively short. Implications for clinical practice could be further elaborated.
- Please include your institutional approval ethics (name, ethics approval number, date approved). Currently is not stated.
Author Response
We sincerely appreciate all reviewer comments and suggestions.
The specific responses to each comment and the specific changes included in the manuscript are in an attached file.
Thank you very much.

Reviewer 2 Report
I have placed my comments in a separate file.

Author Response

(The authors gave the same response as above.)

Reviewer 3 Report
The study can be improved with the following considerations.
1. Clearly describe how the translation works have been done to ensure the linguistic validity.
2. The Zarit scale should be corrected using Zarit Burden Interview. If the authors would like to use an abbreviation, it should be ZBI.
3. The authors mentioned that the aim was "The aim of this study was to focus on the EQ-5D ability to detect differences in HRQOL among caregivers of patients affected by different levels of burden provoked by different types of rare diseases in the European Union, given its increasing use in socioeconomic studies on rare diseases besides its widespread use in economic evaluation studies." However, the entire analyses and results read more like to examine the HRQOL difference between caregivers with different levels of caregiving burden, and to explore the factors associated with HRQOL in these informal caregivers. It does not read like a methodological study to examine the EQ5D capacity.
4. As different algorithms have been developed for EQ5D in different countries, it is unclear to me, which algorithm was used in the present study. Did the authors use the same algorithm for conversion or did they use different algorithms for different countries?
5. I cannot understand why the authors collected a measure of "informal care" (ll148-152). I thought that all the participants were informal caregivers. Also, based on the description on "informal care", it reads like redundant to the other information described in line 156 (i.e., time devoted to care).
6. The authors did the Bonferroni correction. This practice is good. However, the authors should provide details on the Bonferroni correction. That is, what is the adjusted p-value for significance?
7. Tables 1 and 4 are hard to read as the numbers are broken into two lines.
8. Again, I think that the conclusion "In conclusion, we show that the EQ-5D instrument is sensitive to measure differences in HRQOL between caregivers with different levels of care burden." is inappropriate.
9. Lastly, I believe that the present submission can be benefited from professional English editing.
Author Response

(The authors gave the same response as above.)

Round 2
Reviewer 3 Report
The authors have addressed all my prior comments. I have no more comments on this resubmission.
Author Response
We greatly appreciate the reviewer's comments and feedback.